# Baseline Characteristics and Outcomes of Cancer Patients Infected with SARS-CoV-2 in the Lombardy Region, Italy (AIOM-L CORONA): A Multicenter, Observational, Ambispective, Cohort Study

**DOI:** 10.3390/cancers13061324

**Published:** 2021-03-16

**Authors:** Serena Di Cosimo, Barbara Tagliaferri, Daniele Generali, Fabiola Giudici, Francesco Agustoni, Antonio Bernardo, Karen Borgonovo, Gabriella Farina, Giovanna Luchena, Andrea Luciani, Franco Nolè, Laura Palmeri, Filippo Pietrantonio, Guido Poggi, Paolo Andrea Zucali, Emanuela Balletti, Giovanna Catania, Ottavia Bernocchi, Federica D’Antonio, Monica Giordano, Francesco Grossi, Angioletta Lasagna, Nicla La Verde, Mariangela Manzoni, Benedetta Montagna, Angelo Olgiati, Alessandra Raimondi, Irene Rampinelli, Elena Verri, Alberto Zaniboni, Massimo Di Maio, Giordano Beretta, Marco Danova

**Affiliations:** 1Department of Applied Research and Technological Development, Fondazione IRCCS Istituto Nazionale dei Tumori, 20133 Milano, Italy; 2Medical Oncology Unit, ICS Maugeri-IRCCS, 27100 Pavia, Italy; barbara.tagliaferri@icsmaugeri.it (B.T.); antonio.bernardo@icsmaugeri.it (A.B.); emanuela.balletti@icsmaugeri.it (E.B.); 3Department of Medical, Surgery and Health Sciences, University of Trieste, 34123 Trieste, Italy; daniele.generali@gmail.com (D.G.); fgiudici@units.it (F.G.); o.bernocchi@unts.it (O.B.); 4Breast Cancer Unit, Azienda Socio Sanitaria Territoriale di Cremona, 26100 Cremona, Italy; 5Unit of Biostatistics, Epidemiology and Public Health, Department of Cardiac Thoracic Vascular Sciences and Public Health, University of Padova, 35121 Padova, Italy; 6Medical Oncology Department, Fondazione IRCCS Policlinico “San Matteo”, 27100 Pavia, Italy; f.agustoni@smatteo.pv.it (F.A.); a.lasagna@smatteo.pv.it (A.L.); 7Medical Oncology Unit, ASST Bergamo Ovest, 24047 Treviglio, Bergamo, Italy; karen_borgonovo@asst-bgovest.it (K.B.); andrea_luciani@asst-bgovest.it (A.L.); 8Medical Oncology Department, ASST Fatebenefratelli-Sacco, 20146 Milano, Italy; gabriella.farina@asst-fbf-sacco.it; 9Medical Oncology Unit, ASST-Lariana, 22100 Como, Italy; giovanna.luchena@asst-lariana.it (G.L.); monica.giordano@asst-lariana.it (M.G.); 10Division of Urogenital and Head & Neck Tumors, European Institute of Oncology (IEO) IRCCS, 20122 Milano, Italy; franco.nole@ieo.it (F.N.); elena.verri@ieo.it (E.V.); 11Niguarda Cancer Center, Grande Ospedale Metropolitano Niguarda, 20121 Milano, Italy; laura.palmeri@gmail.com; 12Department of Medical Oncology, Fondazione IRCCS Istituto Nazionale dei Tumori, 20133 Milano, Italy; filippo.pietrantonio@istitutotumori.mi.it (F.P.); alessandra.raimondi@istitutotumori.mi.it (A.R.); 13Medical Oncology Unit, Institute of Care Città di Pavia, 27100 Pavia, Italy; guido.poggi@grupposandonato.it (G.P.); benedetta.montagna@grupposandonato.it (B.M.); 14Department of Medical Oncology and Hematology, Humanitas Clinical and Research Center-IRCCS, 20089 Rozzano, Italy; paolo.zucali@cancercenter.humanitas.it (P.A.Z.);; 15Medical Oncology Unit, Azienda Ospedaliera Carlo Poma, 46100 Mantova, Italy; giovanna.catania@asst-mantova.it; 16Department of Medical Oncology, Fondazione IRCCS Ca’Grande Ospedale Maggiore Policlinico, 20162 Milano, Italy; francesco.grossi@policlinico.mi.it; 17Department of Oncology, Luigi Sacco Hospital-ASST Fatebenefratelli Sacco, Via G.B. Grassi 74, 20157 Milan, Italy; nicla.laverde@asst-fbf-sacco.it; 18Department of Medical Oncology, Azienda Ospedaliera “Ospedale Maggiore”, 26013 Crema, Italy; m.manzoni@asst-crema.it; 19Department of Internal Medicine and Oncology, ASST di Pavia, 27100 Pavia, Italy; Angelo_Olgiati@asst-pavia.it; 20Medical Oncology Unit, Multimedica Castellanza, 21010 Varese, Italy; irene.rampinelli@multimedica.it; 21Medical Oncology Unit, Fondazione Poliambulanza, 25124 Brescia, Italy; alberto.zaniboni@poliambulanza.it; 22Medical Oncology Unit, Ordine Mauriziano Hospital, 10128 Torino, Italy; massimo.dimaio@unito.it; 23Department of Medical Oncology, Humanitas Gavezzani, 24125 Bergamo, Italy; giordano.beretta@gavazzeni.it

**Keywords:** cancer, COVID-19, SARS-CoV-2, systemic anti-cancer treatment, mortality

## Abstract

**Simple Summary:**

Cancer patients show an increased vulnerability to SARS-CoV-2 infection and may experience severe COVID-19 complications. AIOM-L Corona aimed to assess the prognostic factors associated with outcomes in 231 cancer patients infected by SARS-CoV-2 between March and September 2020 in Lombardy, the most extensively affected Italian region. A total of 93 events occurred. Known risk factors for mortality in COVID-19 remained significant in the study population. Specifically, age ≥60 years, metastases, dyspnea, desaturation, and interstitial pneumonia were all associated with mortality. Notably, metastatic patients receiving systemic active therapy were less likely to die as compared to untreated counterparts, even after adjusting for other confounding variables (Odds Ratio 0.23, 95%CI 0.11–0.51, *p* < 0.001). While large data sets are needed to confirm these findings, for now, during the COVID-19 pandemic, cancer patients should avoid exposure or increase their protection to SARS-CoV-2 while treatment adjustments and prioritizing vaccination should adequately be considered.

**Abstract:**

Cancer patients may be at high risk of infection and poor outcomes related to SARS-CoV-2. Analyzing their prognosis, examining the effects of baseline characteristics and systemic anti-cancer active therapy (SACT) are critical to their management through the evolving COVID-19 pandemic. The AIOM-L CORONA was a multicenter, observational, ambispective, cohort study, with the intended participation of 26 centers in the Lombardy region (Italy). A total of 231 cases were included between March and September 2020. The median age was 68 years; 151 patients (62.2%) were receiving SACT, mostly chemotherapy. During a median follow-up of 138 days (range 12–218), 93 events occurred. Age ≥60 years, metastatic dissemination, dyspnea, desaturation, and interstitial pneumonia were all independent mortality predictors. Overall SACT had a neutral effect (Odds Ratio [OR] 0.83, 95%Confidence Interval [95%CI] 0.32–2.15); however, metastatic patients receiving SACT were less likely to die as compared to untreated counterparts, after adjusting for other confounding variables (OR 0.23, 95%CI 0.11–0.51, *p* < 0.001). Among cancer patients infected by SARS-CoV-2, those with metastases were most at risk of death, especially in the absence of SACT. During the ongoing pandemic, these vulnerable patients should avoid exposure to SARS-CoV-2, while treatment adjustments and prioritizing vaccination are being considered according to international recommendations.

## 1. Introduction

Coronavirus disease 2019 (COVID-19) is an ongoing pandemic caused by the positive-sense RNA virus named severe acute respiratory syndrome coronavirus 2 (SARS-CoV-2) [1]. The clinical spectrum of COVID-19 is broad, ranging from asymptomatic infection, mild upper respiratory-tract illness, severe viral pneumonia with respiratory failure, to death [2]. Although most people infected with SARS-CoV-2 have mild disease, mortality ranges between 5% and 15% worldwide [3]. On 20 February 2020, Italy was the first European nation to be massively affected by COVID-19, and the biggest cluster of cases occurred in Lombardy, the most populous country region.

Most COVID-19 patients who become severely ill have evidence of underlying conditions such as cardiovascular, liver, and kidney diseases, or malignancies [2,4,5]. Cancer-directed care has been impacted since the early stages of this pandemic, with professional societies advocating for limited access of cancer patients to hospitals, delays in therapy, or the increased usage of targeted and hormonal therapies at the expense of cytotoxic chemotherapy [6,7,8]. These recommendations, based on expert consensus, were issued out of caution given the concern that cancer patients might be at increased risk of SARS-CoV-2 infection, and subsequent increased COVID-19 mortality.

It is now clear that cancer patients infected by SARS-CoV-2 have a poor outcome, with a probability of death as high as 25.6% (95%CI 22.0–29.5%) according to a recent pooled analysis of 52 studies [9]. Whether it is cancer per se that increases the risk of COVID-19 and its complications or rather cancer-associated comorbidities and treatment, remains an issue [10,11]. Therefore, before restrictions to cancer-directed care are implemented in the longer term, it is important to understand which patients are at risk of developing COVID-19 and its complications, with the aim of ensuring protective measures against the pandemic without compromising cancer management. The AIOM-L CORONA study was designed to investigate the main characteristics of cancer patients infected by SARS-CoV-2, to further understand the features of COVID-19 in this patient group, and to elucidate the factors that may affect their clinical outcomes.

## 2. Results

### 2.1. Patient Characteristics and SACT

A total of 231 cancer patients were recorded as being infected by SARS-CoV-2 during the reporting period. The mean age at diagnosis was 68 years (range, 32–90 years) (Table 1), and 113 (48.9%) patients were male. The main cancer diagnoses were 65 breast (28.3%), 42 lung (18.3%), and 38 gastrointestinal (16.5%). Metastatic disease was present in 151 of the 216 (67.1%) patients with solid tumors; bone, lung and liver being the most common sites of spread. Additionally, 144 patients (64.4%) had one or more relevant comorbidities, i.e., disease or medical condition defined *a priori*, and for which the patient was being treated or followed clinical instrumental controls, the most common being hypertension, diabetes, and coronary heart disease (CHD) (Table 1); 73 patients (33.7%) had a present habit or history of smoking.

One hundred and fifty-one patients (66.2%) were receiving SACT, mostly chemotherapy, and 22 (15.2%) had progressive disease at the time of COVID-19 diagnosis (Table 2).

In almost three quarters of all patients, (*n* = 110, 74.5%), the last SACT was with non-curative intent, i.e., first- (*n* = 52, 5.1%), second- (*n* = 35, 23.6%), and third-line treatment or beyond (*n* = 23, 15.5%). The median time from last SACT and COVID-19 diagnosis was 18 days (range, 0–110 days). Among seventy-seven (33.3%) cancer patients not receiving SACT, 29 were in follow-up (5 received last treatment <6 months; 2 received last treatment <9 months; 14 received last treatment for over a year and more; and 8 had still loco-regional disease), 43 were metastatic cases on observation, and 5 were missing. The characteristics of cancer patients not receiving treatment at the time of infection are reported in Appendix A.

### 2.2. COVID-19 Symptoms and Management

A total of 202 (88.2%) patients in this series had clinical symptoms (see Table 3). 

Chest imaging was performed (computed tomography or X-ray) in 194 patients, of whom 83.5% had radiographic evidence of interstitial pneumonia.

Hospital admission occurred in 165 (72%) patients; of which 12 (8%) patients received Intensive Care Unit (ICU)-level care. In univariable analyses, age ≥60 years, male gender, ≥2 comorbidities, ECOG = 1, dyspnea, desaturation, and interstitial pneumonia were all associated with hospital admission (Table 4). Male gender, desaturation, and interstitial pneumonia remained independent risk factors for hospital admission at multivariable analyses.

COVID-19-directed therapies included hydroxychloroquine in 10 patients (6.1%), antivirals in 71 (43.6%), antibiotics in 16 (9.8%), supplemental oxygen in 147 (90.2%), and non-invasive respiratory support in 19 (Continuous Positive Airway Pressure [CPAP], 11.7%). Sixty-one (26.9%) patients did not receive any of these therapies; most being female (41, 67.2%) and 28 with breast cancer (45.9%), 49 on SACT (81.7%), 12 presenting with dyspnea (21%), and only 3 desaturation (5.3%) (Appendix A).

### 2.3. Survival Outcome

Of the 93 (41.8%) deaths recorded during a median follow-up of 138 days (range 12–218), 81 (36.5%) occurred within 30 days from the initial diagnosis of COVID-19. Case fatality rates are likely to be underestimated as live status was unknown for 9 cases at the time of this analysis. Notably, 16 (12.2%) patients presented with progressive disease at the first follow-up visit after recovery from COVID-19. Table 5 describes the univariable analysis of baseline characteristics as predictors of 30-day mortality. Age ≥60 years, metastatic disease, dyspnea, desaturation, interstitial pneumonia, ≥2 comorbidities, ECOG ≥ 1, and a current diagnosis of lung cancer were all associated with increased risk of mortality. The first five variables remained significantly associated with worse outcome also at multivariable analysis. Notably, metastatic cancer patients receiving SACT had a decreased risk of death as compared with untreated counterparts. Even when adjusted for age and gender, having received anti-cancer directed therapy impacted 30-day mortality (odds ratio [OR] = 0.23; 95%CI, 0.11–0.51; *p* ≤ 0.001). Treatment with antivirals was an independent favorable prognostic factor (OR = 0.48; 95%CI, 0.24–0.94; *p* = 0.03).

## 3. Discussion

Different international registries have been published to identify the clinical characteristics of cancer patients with severe COVID-19 [12,13,14,15,16,17,18,19,20]. Overall, the data suggest that cancer patients are susceptible to SARS-CoV-2 infection, with prevalence rates varying across different studies between 0.1% and 2.5%, and vulnerable to COVID-19 complications, with case fatality rates between 5% and 61%. This ill-defined figure reflects the heterogeneity of case series analyzed, with most reports referring to retrospective case series, including exclusively hematologic malignancies, and/or missing important solid tumor information, such as disease stage, and prior or current SACT. It has been reported that the susceptibility to the infection and the characteristics of the COVID-19 are variable depending on the tumor type, stage and active treatment, but it remains unclear as to whether the risk factors previously described as associated with adverse COVID-19 outcome in the general population are still important within the cancer patient population. The AIOM-L CORONA aimed to evaluate the relationship between individual, disease, and treatment characteristics with outcome by taking into account various confounding factors. Several key findings with direct clinical relevance were found.

Henceforth, pre-existing diagnoses of hypertension, diabetes, CHD, atrial fibrillation as well as chronic obstructive pulmonary disease (COPD) have been associated with an increased risk of hospitalization and death in the general population [4,5]. The distribution of these conditions in our cancer cohort reflected that of the whole Italian population, i.e., prevalence of hypertension, diabetes, and CHD of 31%, 13%, and 2.5–20%, respectively [21]. Of note, patients with at least two comorbidities were approximately three-fold more likely to require hospitalization, and to die within 30 days from infection. However, this excess of risk disappeared after adjusting for age and gender, suggesting that it was due to patients being men and old, rather than to an inherent comorbidity independent effect. Interestingly, in our cancer cohort, two third of patients succumbing to COVID-19 were male with a median age of 74 years. This is 10 years older than the entire cohort, which is consistent with rapidly accumulating evidence that gender and age play a major role in COVID-19 severity [22,23].

In contrast with data from a European [24], and a Chinese study [12], we found that patients with lung cancer were relatively under-represented being less than a quarter of the study population. The reasons for this are unclear due to the number of patients involved and stochastic effects. However, this finding is in line with the United Kingdom (UK) Coronavirus Cancer Monitoring Project (UKCCMP) [19,20], which recently reported that the diagnosis of lung cancer did not impact the risk of COVID-19 infection. Regardless of their a priori susceptibility to COVID-19 infection, in our study patients with lung cancer had a significantly worse risk of mortality if infected, at least at univariable analysis. This is reflected in work elsewhere [14].

There has been much discussion on whether being on SACT, specifically chemotherapy, has an adverse effect on outcomes of COVID-19 positive patients. A study from China with 105 COVID-19 positive patients suggested that chemotherapy is associated with increased mortality [12]. However, this has been disputed by the UKCCMP suggesting that recent SACT regardless of treatment modality is not associated with worse outcomes in COVID-19 positive patients [20]. Our study showed that SACT is not a risk factor for 30-day mortality in cancer patients with COVID-19. Therefore, withholding or discontinuing SACT out of fear of COVID-19 might not be warranted. Furthermore, SACT appeared to reduce the risk of mortality in patients with metastatic disease, and this remained true even after adjusting for age and gender. It can be hypothesized that SACT, especially chemotherapy, may affect humoral and cellular immune function of treated cases, exerting some type of anti-inflammatory effect (reviewed in [25]). However, patients receiving SACT did not suffer a less aggressive COVID-19 trajectory in our cohort, as they experienced similar rates of hospital and ICU admissions as patients not receiving any SACT (Appendix A). Additionally, metastatic patients not receiving SACT were more likely to have lung cancer and hematologic malignancies, and to present with worse performance status (Appendix A), which speak to their general frailty and possibly increased vulnerability to COVID-19. Although disease site and ECOG were not prognostic factors in the analyses we reported, we recognize that our patient cohort was small, and therefore these findings are only preliminary and should not be over interpreted. 

While we did not aim to specifically examine efficacy of treatment options for cancer patients with symptomatic COVID-19, we acknowledge two valuable findings. First, we found a significant reduction of mortality in patients receiving antiviral-based therapy. These findings are consistent with the national and international recommendations for the treatment of hospitalized patients requiring oxygen supplementation [26,27]. Second, just 8% of hospitalized patients were admitted to ICU. In Italy, patients with COVID-19 are almost solely admitted to the ICU when mechanical ventilation is required, yet the rate of ICU admissions in our study was one third that reported in publicly available databases of COVID-19 positive cases [28]. Discussing treatment restrictions with cancer patients was a well-established practice even before COVID-19, especially for cases with incurable cancers. However, SACT has changed the life expectancy of patients with metastatic disease, and therefore the decision to use intensive care should be made in a multidisciplinary setting. Biases that limit access to treatment of patients with metastatic disease should be avoided because although most are treated with non-curative intent, treatment still aims to control the disease long term [29].

We recognize several limitations in our study. First, the study was only conducted by medical oncologists, resulting in a potential selection bias with under-estimation of specific groups of patients, for instance, those who did not attend the hospital, or died in an out-of-hospital setting. Second, the study could not estimate the prevalence and incidence of COVID-19 among cancer patients in the Lombardy region because it failed to capture most asymptomatic and/or mildly symptomatic cases not subject to testing. The optimal situation would have been to have a representative sample of the most common cancer types tested for SARS-CoV-2 infection to better understand COVID-19 epidemiology, but this was not possible at the height of the pandemic due to limited testing resources. Furthermore, the scientific community advised asymptomatic patients, including those with cancer, to seek medical care only if strictly necessary. We note, however, that no such data sets are yet available for any specific malignancy, as acknowledged also by a recently published Italian study addressing the incidence of COVID-19 among cancer patients [30]. Finally, this study intentionally did not comment on the optimal management of COVID-19, nor analyzed data in this regard, other than confirming the efficacy of antivirals, because it took place at a time when anti-COVID-19 practices were heterogeneous and evolving as rapidly as the pandemic.

## 4. Methods

This was a multicenter, observational, ambispective, cohort study supported by AIOM (Associazione Italiana di Oncologia Medica). The study started on 15 May 2020 with the intended participation of 26 centres in the Lombardy region. Investigators at each center used a standardized case report form to collect data on cancer patients (i.e., males or females of any age, with any active malignancy of any histology/stage, including those on systemic anti-cancer treatment [SACT], i.e., any therapy given within 4 weeks before the diagnosis of infection, or in clinical follow-up after neo-/adjuvant treatment), diagnosed with COVID-19 based on the presence of SARS-CoV-2 RNA confirmed by reverse transcriptase quantitative polymerase chain reaction during the first 6 months of the pandemic. Therefore, the prospective portion of AIOM-L CORONA included longitudinal cases of cancer patients with COVID-19 starting from the date of study initiation at each participating center until study end, set as September 30, 2020 for all participants. The retrospective portion of AIOM-L CORONA included cases retrieved at each participating center from 1 March 2020 and monitored until study end. Data collected included demographics, baseline characteristics, predefined comorbidities, SACT, COVID-19 signs, symptoms, and management, including antiviral therapy (i.e., remdesevir 200 mg i.v., day 1; 100 mg i.v., daily from day 2 during a minimum of 5 to a maximum of 10 days), and clinical outcomes, i.e., hospital admission, and vital status. AIOM-L CORONA was Institutional Review Board approved and conducted in accordance with the Declaration of Helsinki.

## 5. Statistical Analysis

Descriptive statistics of patient demographics (i.e., age, gender) and clinical characteristics (i.e., comorbidities, SACT type) were reported as frequencies (proportions) for categorical variables, and median (range max-min) continuous variables. Utilizing Cox regression, univariable analyses were performed to evaluate the relation between baseline characteristics, and clinical outcome (i.e., hospital admission, and death from any cause, in or out of hospital within 30 days from the initial diagnosis of infection (30-day mortality). Significant predictors from univariable analyses (*p* < 0.05) were included in a multivariable model. All the analyses were exploratory with no predefined statistical hypothesis to test. To ensure sufficient precision in descriptive outcomes, considering 26 participating centers, a study drop out of 50% given the challenge of engaging clinicians during a pandemic emergency, and a prevalence of SARS-CoV-2-infection among cancer patients of approximately 1% [11], a sample size of 212 patients was expected to produce a Confidence Interval (CI) of ±3.5% for estimates of proportion. Statistical analyses were performed using statistical software R (version 4.0.2, Free Software Foundation’s GNU General Public License, www.gnu.org (accessed on 16 October 2020)). The database was locked on 16/10/2020 for analyses.

## 6. Conclusions

The AIOM-L CORONA is the first large report of COVID-19 among patients with cancer in the first and most affected Italian region of Lombardy. In the study population, we found a similar, if not numerically superior, mortality rate compared with other population-based datasets studying similar patient metrics. Known risk factors for mortality in COVID-19, including age and male gender, remain significant in this cancer patient population. Patients who received SACT do not have a worse prognosis regardless of disease stage. These findings suggest that postponing treatment for fear ofinfection between treatment cycles, appears unwarranted given SACT does not compromise prognosis once infection occurs. In contrast, metastatic patients receiving SACT had a reduced risk of mortality as compared with untreated counterparts. Large data sets are needed to validate our initial findings on the protective role of chemotherapy, and especially for patient management after the diagnosis of COVID-19, which remains yet to be defined. For now, during the COVID-19 pandemic, these vulnerable patients should avoid exposure or increase their protection to SARS-CoV-2 while treatment adjustments should be considered on value-based prioritization and clinical cogency of interventions as recently recommended by international expert consensus [7]; finally, in line with the WHO principles and objectives aiming to reduce deaths and disease burden, cancer patients deserve an additional priority for vaccination unless limitations exist [31,32].

## Figures and Tables

**Table 1 cancers-13-01324-t001:** Demographic and baseline clinical characteristics of SARS-CoV-2-infected cancer patients.

Characteristics	Patients (*N* = 231)
**Mean (Standard Deviation), years**	68 (11)
**Median age (Range), years**	68 (32–90)
**Male sex (N, %)**	113 (48.9%)
**Smoking History (N, %)**	
Never smoker	143 (66.2%)
Active smoker	48 (22.2%)
Past smoker	25 (11.6%)
Subtotal	216
Unknown	15 (6.5%)
**Comorbidity (N, %)**	
None	82 (35.6%)
1	96 (41.7%)
≥2	52 (22.6%)
Subtotal	230
Unknown	1 (0.4%)
**Type of Comorbidity ^a^ (N, %)**	
Hypertension	97 (42.2%)
Diabetes	43 (18.7%)
Coronary heart disease	36 (15.7%)
Arrhythmia	25 (10.9%)
Chronic pulmonary disease	19 (8.3%)
**ECOG Performance Status (N, %)**	
0–1	145 (91.2%)
2–3	14 (8.8%)
Subtotal	159
Unknown	72 (31.2%)
**Tumour diagnosis (N, %)**	
Breast	65 (28.3%)
Lung	42 (18.3%)
GI-no colorectal	38 (16.5%)
Colorectal	26 (11.3%)
Genitourinary tract cancer	25 (10.9%)
Hematological	14 (6.1%)
Head and Neck	9 (3.9%)
Gynecological	5(2.2%)
Brain	3 (1.3%)
Melanoma	3 (1.3%)
Subtotal	230
Unknown	1 (0.4%)
**Stage (N, %)**	
I	24 (10.7%)
II	21 (9.3%)
III	29 (12.9%)
IV	151 (67.1%)
Subtotal	225
Unknown (N, %)	6 (2.6%)
**Site of Metastasis for IV stage patients ^b^ (N, %)**	
Lung	53 (36.3%)
Bone	53 (36.3%)
Liver	46 (31.5%)
Lymph nodes	37 (25.3%)
Other	32 (21.9%)
Brain	21 (14.4%)
Peritoneum	6 (4.1%)
Soft Tissue	3 (2.1%)
Subtotal IV stage patients	146
Unknown	1 (0.7%)
**Site of metastases ≥3 for IV stage patients (N, %)**	
Yes	27 (18.5%)
No	119 (81.5%)
Subtotal	146
Unknown	1 (0.7%)

^a^ patients can have more than one comorbidity; ^b^ patients can have more than one site.

**Table 2 cancers-13-01324-t002:** Anti-cancer treatment in SARS-CoV-2-infected cancer patients.

Treatment Characteristics	Patients (*N* = 231)
**Active Anticancer Treatment (N, %)**	
Yes	151 (66.2%)
No	77 (33.8%)
Subtotal	228
Unknown (N, %)	3 (1.3%)
**Type of Anticancer Treatment (N, %) ^a^**	
Chemotherapy	79 (52.7%)
Immunotherapy	14 (9.3%)
Biological therapy	42 (28.0%)
Ormonotherapy	32 (21.3%)
Subtotal	148
Unknown (N, %)	3 (2.0%)
**Type of Anticancer treatment (N, %)**	
Neoadjuvant	7 (4.7%)
Adjuvant	31 (20.9%)
1st line	52 (35.1%)
2nd line	35 (23.6%)
≥3rd line	23 (15.5%)
Subtotal	148
Unknown (N, %)	3 (2.0%)
**No Anticancer Treatment (N, %)**	
Follow-up after therapy	29 (40.3%)
Temporary treatment discontinuation in metastatic patients	43 (59.7%)
Subtotal	72
Unknown (N, %)	5 (6.5%)
**Time between last therapy and COVID-19 (days)**	
Median (Min, Max)	18 (0–110)
Subtotal	98
Unknown	53 (35.1%)

^a^ patients can have more than one treatment.

**Table 3 cancers-13-01324-t003:** Symptoms, treatments, and clinical outcomes of SARS-CoV-2-infected cancer patients.

Characteristics (N, %)	Patients (*N* = 231)
**Symptoms and signs**	
Yes	202 (88.2%)
No	27 (11.8%)
Unknown	2 (0.9%)
**Type of Symptoms**	
Fever	156 (68.7%)
Dyspnea	123 (54.4%)
Desaturation (SpO2 ≤ 93%)	120 (54.1%)
Cough	96 (42.9%)
**Interstitial pneumonia**	
Yes	162 (83.5%)
No	32 (16.5%)
Unknown	37 (16.0%)
**Covid-19 Therapy**	
Yes	166 (73.1%)
No	61 (26.9%)
Unknown	4 (1.7%)
**Type of anti-COVID-19 therapy ^a^**	
Supplemental Oxygen ^	147 (90.2%)
Antiviral Treatment ^^	71 (43.6%)
CPAP	19 (11.7%)
Antibiotic Treatment	16 (9.8%)
Hydroxychloroquine	10 (6.1%)
Other not specified	3 (1.8%)
Unknown	3 (1.8%)
**Progression Disease at COVID-19 diagnosis**	
Yes	22 (15.2%)
No	123 (84.8%)
Unknown	86 (37.2%)
**Oncological Progression Disease after COVID-19**	
Yes	16 (12.2%)
No	115 (87.8%)
Unknown	100 (43.3%)
**Ordinary Hospitalization**	
Yes	165 (72.1%)
No	64 (27.9%)
Unknown	2 (0.9%)
**Admission to Intensive Care Unit after Ordinary Hospitalization**	
Yes	12 (8.0%)
No	138 (92.0%)
Unknown	15 (9.1%)
**Total hospital length of stay (hospitalized patients only)-days**	
Median (Min-Max)	12 (1–137)
Subtotal	165
Unknown	75 (45.5%)
**Mortality within 30 days of COVID-19 diagnosis**	
Yes	81 (36.5%)
No	141 (66.5%)
Unknown	9 (3.9%)

^a^ patients can receive more than one therapy; ^ Supplemental oxygen included conventional oxygen therapy, i.e., Venturi Mask or mask with or without a reservoir bag; or high flow nasal cannula as per local practice; ^^ Remdesivir.

**Table 4 cancers-13-01324-t004:** Logistic regression analysis of potential factors associated with hospital admission (*n* = 165) for SARS-CoV-2-infected cancer patients.

Factor	Univariable OR (95%CI)	*p*-Value	Multivariable OR	*p*-Value
			(95%CI)	
**Age (Years)**				
<60	1.00 (Reference)		1.00 (Reference)	
≥60	2.21 (1.17–4.20)	0.015	1.06 (0.40–2.83)	0.9
**Gender**				
Female	1.00 (Reference)		1.00 (Reference)	
Male	2.09 (1.15–3.79)	0.015	2.57 (1.01–6.59)	0.04
**Smoking History**				
Never smoker	1.00 (Reference)			
Active smoker	2.00 (0.89–4.49)	0.12		
Past smoker	2.49 (0.81–7.67)	0.09		
**Comorbidity**				
None	1.00 (Reference)		1.00 (Reference)	
1	0.88 (0.47–1.66)	0.69	0.61 (0.24–1.58)	0.31
≥2	2.87 (1.14–7.24)	0.026	4.81 (0.82–28.1)	0.08
**ECOG**				
0–1	1.00 (Reference)			
2–3	0.59 (0.19–1.81)	0.36		
**Tumor**				
No lung Cancer	1.00 (Reference)			
Lung Cancer	2.21 (0.93–5.27)	0.07		
**Stage**				
I–III	1.00 (Reference)			
IV	1.15 (0.61–2.16)	0.67		
**Active Anticancer Treatment in Metastatic Patients**				
No	1.00 (Reference)		1.00 (Reference)	
Yes	0.11 (0.03–0.38)	<0.001	0.12 (0.03–0.40) ^a^	<0.001
**Active Anticancer Treatment in non-metastatic Patients**				
No				
Yes	0.45 (0.14–1.47)	0.19		
**Line of Therapy**				
**Neo-/Adjuvant**	1.00 (Reference)			
**≥1st line**	0.71 (0.32–1.55)	0.39		
**Type of Drug**				
Chemotherapy	1.00 (Reference)			
Chemotherapy+Biological	0.50 (0.14–1.83)	0.3		
Biological	0.69 (0.29–1.68)	0.42		
Hormonotherapy	0.45 (0.18–1.12)	0.09		
Immunotherapy	0.67 (0.19–2.29)	0.52		
**Duration of therapy (days)**	1.00 (0.998–1.00)	0.47		
**Last treatment within 4 weeks before infection**				
No	1.00 (Reference)			
Yes	0.93 (0.37–2.34)	0.89		
**Progression disease at COVID-19**				
No	1.00 (Reference)			
Yes	5.25 (1.17–23.50)	0.03	no for missing data more than 10%	
**Symptoms**				
No	1.00 (Reference)			
Yes	4.26 (1.83–9.88)	<0.001	no for collinearity	
**Fever**				
No	1.00 (Reference)			
Yes	1.83 (1.00–3.37)	0.05		
**Cough**				
No	1.00 (Reference)			
Yes	0.88 (0.49–1.58)	0.67		
**Dyspnea**				
No	1.00 (Reference)		1.00 (Reference)	
Yes	4.79 (2.54–9.03)	<0.001	0.90 (0.30–2.67)	0.0.85
**Desaturation**				
No	1.00 (Reference)		1.00 (Reference)	
Yes	14.20 (6.47–31.00)	<0.001	17.5 (5.23–58.8)	<0.001
**Interstitial pneumonia**				
No	1.00 (Reference)		1.00 (Reference)	
Yes	7.33 (3.24–16.60)	<0.001	2.95 (0.98–8.84)	0.05

^a^ adjusted for age and gender.

**Table 5 cancers-13-01324-t005:** Logistic regression analysis of potential prognostic factors associated with 30-day mortality (*n* = 81) for SARS-CoV-2-infected cancer patients.

Factor	Univariable OR (95%CI)	*p*-Value	Multivariable OR (95%CI)	*p*-Value
**Age (Years)**				
<60	1.00 (Reference)			
≥60	4.62 (2.05–10.40)	<0.001	2.92 (1.14–7.50)	0.026
**Gender**				
Female	1.00 (Reference)			
Male	1.67 (0.97–2.91)	0.06		
**Smoking History**				
Never smoker	1.00 (Reference)			
Active smoker	2.17 (1.10–4.27)	0.03	1.68 (0.74–3.77)	0.21
Past smoker	2.07 (0.85–5.08)	0.11	2.29 (0.80–6.56)	0.12
**Comorbidity**				
None	1.00 (Reference)			
1	1.23 (0.64–2.37)	0.54	1.23 (0.57–2.70)	0.6
≥2	3.18 (1.51–6.71)	0.002	1.96 (0.79–4.87)	0.15
**ECOG**				
0–1	1.00 (Reference)		no for missing data more than 10%	
2–3	5.68 (1.65–19.5)	0.006		
**Tumor**				
No Lung Cancer	1.00 (Reference)			
Lung Cancer	2.14 (1.06–4.31)	0.03	0.92 (0.39–2.22)	0.84
**Stage**				
I–III	1.00 (Reference)			
IV	1.88 (1.01–3.49)	0.047	2.09 (0.99–4.44)	0.05
**Active Anticancer Treatment in metastatic patients**				
No	1.00 (Reference)		1.00 (Reference)	
Yes	0.22 (0.10–0.47)	<0.001	0.23 (0.11–0.51) ^a^	<0.001
**Active Anticancer Treatment in no-metastatic patients**				
No	1.00 (Reference)			
Yes	0.54 (0.17–1.68)	0.28		
**Line of Therapy**				
Neo-/adjuvant	1.00 (Reference)			
≥1st Line	1.93 (0.76–4.85)	0.16		
**Type of Therapy**				
Chemotherapy	1.00 (Reference)			
Chemotherapy+Biological	0.00 (0.00–NA)	0.99		
Biological	0.75 (0.28–1.95)	0.55		
Hormonotherapy	0.47 (0.16–1.40)	0.17		
Immunotherapy	2.05 (0.59–7.12)	0.26		
**Duration of therapy (days)**	1.00 (0.99–1.00)	0.61		
**Last treatment within 4 weeks before infection**				
No	1.00 (Reference)			
Yes	0.83 (0.32–2.15)	0.7		
**Progression disease at COVID-19 Diagnosis**			no for missing data more than 10%	
No	1.00 (Reference)			
Yes	2.10 (0.83–5.33)	0.12		
**Progression disease after COVID-19**			no for missing data more than 10%	
No	1.00 (Reference)			
Yes	0.51 (0.14–1.89)	0.31		
**Symptoms**				
No	1.00 (Reference)			
Yes	2.71 (0.98–7.48)	0.06		
**Fever**				
No	1.00 (Reference)			
Yes	1.13 (0.62–2.05)	0.69		
**Cough**				
No	1.00 (Reference)			
Yes	0.57 (0.33–1.01)	0.06		
**Dyspnea**				
No	1.00 (Reference)			
Yes	2.81 (157–2.04)	<0.001	1.85 (0.81–4.23)	0.14
**Desaturation**				
No	1.00 (Reference)			
Yes	2.90 (161–5.23)	<0.001	1.62 (0.71–3.73)	0.25
**Interstitial pneumonia**				
No	1.00 (Reference)			
Yes	1.32 (0.57–3.05)	0.52		
**Type of anti-COVID-19 Treatment**				
No Antiviral	1.00 (Reference)		1.00 (Reference)	
Antiviral	0.43 (0.22–0.43)	0.01	0.48 (0.24–0.94) ^a^	0.03

^a^ adjusted for age and gender.

## Data Availability

The data presented in this study are available on request from the corresponding authors. The data are not publicly available due to due to privacy restrictions.

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
