# Peer review of "Baseline Characteristics and Outcomes of Cancer Patients Infected with SARS-CoV-2 in the Lombardy Region, Italy (AIOM-L CORONA): A Multicenter, Observational, Ambispective, Cohort Study"

_cancers, 2021, doi:10.3390/cancers13061324_

Round 1

Reviewer 1 Report

We praise the authors' work. Nevertheless the findings of the study are hardly original in light of currently available data from larger cohorts. I have some corrections:

-Abstract:  'Patients with metastatic cancer are at increased risk of death after COVID-19' Compared to what? There is no reference for this sentence in the text. 'During the ongoing pandemic, these vulnerable patients should avoid exposure to SARS-CoV-2, while treatment adjustments and prioritizing vaccination are being considered.' This is also not referred in the text (only in the conclusion) not there is a basis for this affirmation which is very generic.

-Nomenclature needs to be unified across the article (the disease is referred as either COVID, Covid 19, COVID19 or COVID-19, especially in table 3… and the virus as SARS-CoV-2 or SARS-CoV2). Use either COVID-19 or Covid-19, and SARS-CoV-2.

-Information is redundant in the text and the tables.

-Table 2: Correct Ormonotherapy (Hormonotherapy).

-Table 3: add type of oxygen delivery (low flow, high flow, high flow nasal cannula, invasive ventilation) if available.

-Line 202: The sentence in line 202 ‘SACT decreased the risk of mortality by 80%’ is incorrect since it may suggest casuality. It would be more correct to say ‘SACT was associated with a lower risk of mortality’.  As the authors hypothetize this is probably due to confounders such the fact that patients with poorer PS or more advanced disease may not be receiving SACT and has been described in other cohorts (references could be added). Same for line 254.

Line 258: How long after finishing adjuvant therapy are patients considered candidates to be included in the study? Since these patients constituted 40% of the cohort it would be interesting to know if these are, for example, patients with recent treatment of >6 months. In cases with more remote treatments, patients can hardly be considered ‘cancer patients’ since the effect of chemotherapy or cured cancer on immunity is neligible. This information should be made available in results and conclusions.

-Methods: Define antiviral therapy

-Conclusion: 'For now, during the COVID-19 pandemic, these vulnerable patients should avoid exposure or increase their protection to SARS-CoV-2 while treatment adjustments and prioritizing vaccination should adequately 259be considered' From where does this sentence derive in light of the article's findings? What does 'increase their protection' mean?

Author Response

REVIEWER#1 (Comments to the Author):

We praise the authors' work. Nevertheless the findings of the study are hardly original in light of currently available data from larger cohorts. I have some corrections:

  1. Reviewer Comment/Question 1:

Abstract: 'Patients with metastatic cancer are at increased risk of death after COVID-19' Compared to what?

Authors reply to the Comment/Question 1:

We would like to thank the Referee for this point. In this regard, we changed the sentence in the abstract as following: “Among cancer patients infected by SARS-CoV-2, those with metastases were most at risk of death”

  1. Reviewer Comment/Question 2:

There is no reference for this sentence in the text. 'During the ongoing pandemic, these vulnerable patients should avoid exposure to SARS-CoV-2, while treatment adjustments and prioritizing vaccination are being considered.' This is also not referred in the text (only in the conclusion) not there is a basis for this affirmation which is very generic.

Authors reply to the Comment/Question 2:

We agree with the Referee that this information was missing in the previous version of the manuscript and we revised accordingly. Specifically, we modified the last sentence of the abstract adding the reference to international guidelines and the main text (in the conclusion) as following For now, during the COVID-19 pandemic, these vulnerable patients should avoid exposure or increase their protection to SARS-CoV-2 while treatment adjustments should be considered on value-based prioritisation and clinical cogency of interventions as recently recommended by international guidelines [31]; finally, in line with the WHO principles and objectives aiming to reduce deaths and disease burden, cancer patients deserve an additional priority for vaccination unless limitations exist [32]. New references refer to the tiered approach in cancer patient management during the pandemic through three levels of priority, namely: level 1 (high priority intervention), 2 (medium priority) and 3 (low priority) - defined according to the Cancer Care Ontario, Huntsman Cancer Institute and ESMO-Magnitude of Clinical Benefit Scale (ESMO-MCBS) criteria, incorporating information on value-based prioritisation and clinical cogency of interventions (Curigliano G et al. 2020); and to the recently published guidelines for vaccination of cancer patients (Garassino M et al. 2021).

  1. Reviewer Comment/Question 3:

Nomenclature needs to be unified across the article (the disease is referred as either COVID, Covid 19, COVID19 or COVID-19, especially in table 3… and the virus as SARS-CoV-2 or SARS-CoV2). Use either COVID-19 or Covid-19, and SARS-CoV-2.

Authors reply to the Comment/Question 3:

We thank the Referee for raising the issue of nomenclature consistency. We used SARS-CoV-2 for virus, and COVID-19 for disease.

  1. Reviewer Comment/Question 4:

Information is redundant in the text and the tables.

Authors reply to the Comment/Question 4:

Tables and text were revised to reduce at minimum redundancy. Please, refer to specific comments in the tracked version.

  1. Reviewer Comment/Question 5:

Table 2: Correct Ormonotherapy (Hormonotherapy).

Authors reply to the Comment/Question 5:

Fixed.

  1. Reviewer Comment/Question 6:

Table 3: add type of oxygen delivery (low flow, high flow, high flow nasal cannula, invasive ventilation) if available.

Authors reply to the Comment/Question 6:

The caption of Table 3 now includes the type of supplemental oxygen, i.e. conventional oxygen therapy (Venturi Mask or mask with or without a reservoir bag); or high flow nasal cannula, as per local practice.

  1. Reviewer Comment/Question 7:

Line 202: The sentence in line 202 ‘SACT decreased the risk of mortality by 80%’ is incorrect since it may suggest casuality. It would be more correct to say ‘SACT was associated with a lower risk of mortality’. As the authors hypothetize this is probably due to confounders such the fact that patients with poorer PS or more advanced disease may not be receiving SACT and has been described in other cohorts (references could be added). Same for line 254.

Authors reply to the Comment/Question 7:

Accordingly, we modified both sentences in the text as suggested by the Referee. As compared to patients on active therapy, untreated cases were more likely to present with ECOG 2-3, i.e. 5/33 (15%) versus 6/75 (8%), p= 0.26, lung cancer, i.e. 14/43 (32.5%) versus 21/108 (19.4%), p= 0.08, and haematological disease, i.e. 6/43 versus 5/108, p= 0.04 (this information has been introduced in the caption of Supplementary Table S2, which already reported in bold the main features of untreated patients). Notably, ECOG and site of disease were not significant prognostic factors in the logistic regression models for hospitalization and death (Tables 4-5, main text). However, we cannot exclude (or state) the presence of other confounding factors. Considering the length of the text, at the maximum limit, we think that this figure has already been sufficiently focused in the discussion (please, refer to highlighted text).

  1. Reviewer Comment/Question 8:

Line 258: How long after finishing adjuvant therapy are patients considered candidates to be included in the study? Since these patients constituted 40% of the cohort it would be interesting to know if these are, for example, patients with recent treatment of >6 months. In cases with more remote treatments, patients can hardly be considered ‘cancer patients’ since the effect of chemotherapy or cured cancer on immunity is neligible. This information should be made available in results and conclusions.

Authors reply to the Comment/Question 8:

We thank the Referee for stressing this point, and for giving us the possibility to better explain the patient characteristics and the objectives of AIOM-L CORONA study. Among the 29 follow-up patients mentioned by the Referee, 5 received treatment <6 months; 2 received treatment <9 months; 8 had loco-regional disease (3 biliary tract cancer, 2 gastro-intestinal, 2 genitourinary, 1 lung cancer cases); and 14 received last treatment for over a year and more (these data were reported in the current text). We recognize that these latter patients are different from those on recent treatment or with disease. However, we cannot help but consider that these cases were were considered worthy of clinical-instrumental follow-up by their treating oncologist and therefore considered at risk of relapse or late treatment toxicity. AIOM-L CORONA was not designed to collect many details because the study was performed in an emergency setting and therefore we cannot understand why patients were in follow-up and with what kind of schedule. In addition this population was heterogenous for cancer type (Table S1). Therefore, we believe that emphasizing this subgroup, and specifically the14 patients far from treatment, does not add much information to the manuscript. Considering also that the focus of the article is on patients being treated for disease, and especially for metastases, we would avoid to over-interpret the aforementioned data. In fact, should the above commented data be included in the discussion, this could make the reading of the manuscript more cumbersome and take us out of context.

  1. Reviewer Comment/Question 9:

-Methods: Define antiviral therapy

Authors reply to the Comment/Question 9:

Absolutely. We included the following information in the Methods “including antiviral therapy (i.e., remdesevir 200 mg i.v., day 1; 100 mg i.v. daily from day 2 during a minimum of 5 to a maximum of 10 days),” and in the caption of Table 6.

  1. Reviewer Comment/Question 10:

Conclusion: 'For now, during the COVID-19 pandemic, these vulnerable patients should avoid exposure or increase their protection to SARS-CoV-2 while treatment adjustments and prioritizing vaccination should adequately 259be considered' From where does this sentence derive in light of the article's findings? What does 'increase their protection' mean?

Authors reply to the Comment/Question 10:

Please refer to the answer to the comment/question 2.

Reviewer 2 Report

Authors Di Cosimo et al. define the impact of COVID-19 on cancer patients in the Lombardy region of Italy. They identified several factors on multivariable analysis that predict for need of COVID-19-related hospitalization or 30-day mortality. The paper is well written with appropriate statistical analysis. Several small issues need to be addressed, but otherwise the paper is fit for publication.

Line 101, 102, 104, 105: please list the total number of patients next to the %. Please correct this in other parts of the paper.

Line 104: please define “relevant comorbidities”. Were they defined a priori?

Table 1: for the subtitle “Site of metastases >=3 for IV stage patients”, it is unclear where the * is directing the reader for footnotes.

Table 2: subtitle “Type of Anticancer Treatment”. I presume “Ormonotherapy” is a typo for “Hormone therapy”?

Line 128: please define “desaturation” based on oxygen saturation percent.

Tables 4 and 5: Under the multivariable OR column, why are some cells blank while others have “//” present? Further, several * symbols are present in the multivariable OR column (e.g. Table 4 row Active Anticancer Treatment in Metastatic Patients, Table 5 row Type of COVID treatment”) but it is unclear where the * is directing the reader for footnotes.

Line 190: please define UKCCMP

Lines 202-204: The language in the following sentence is too strong: “Furthermore, SACT de- creased the risk of mortality by 80% in patients with metastatic disease, and this remained true even after adjusting for age and gender.” This study is underpowered to draw this conclusion. Please rephrase. This is similar in the conclusion paragraph lines 254-255. Furthermore, there is high likelihood that metastatic patients not on SACT have poor performance status and/or refractory disease that precludes further SACT, which speaks to their general frailty and possibly increased vulnerability to COVID-19. If this analysis has not been performed, it should at least be mentioned in the discussion.

Author Response

Please see attachment for details

REVIEWER#2 (Comments to the Author):

Authors Di Cosimo et al. define the impact of COVID-19 on cancer patients in the Lombardy region of Italy. They identified several factors on multivariable analysis that predict for need of COVID-19-related hospitalization or 30-day mortality. The paper is well written with appropriate statistical analysis. Several small issues need to be addressed, but otherwise the paper is fit for publication.

  1. Reviewer Comment/Question 1:

Line 101, 102, 104, 105: please list the total number of patients next to the %. Please correct this in other parts of the paper.

Authors reply to the Comment/Question 1:

We modified accordingly, number are reported before %.

  1. Reviewer Comment/Question 2:

Line 104: please define “relevant comorbidities”. Were they defined a priori?

Authors reply to the Comment/Question 2:

Relevant comorbidities included disease or medical condition for which the patient was being treated or followed clinical instrumental controls and were defined a priori.

  1. Reviewer Comment/Question 3:

Table 1: for the subtitle “Site of metastases >=3 for IV stage patients”, it is unclear where the * is directing the reader for footnotes.

Authors reply to the Comment/Question 3:

Fixed.

  1. Reviewer Comment/Question 4:

Table 2: subtitle “Type of Anticancer Treatment”. I presume “Ormonotherapy” is a typo for “Hormone therapy”?

Authors reply to the Comment/Question 4:

Fixed.

  1. Reviewer Comment/Question 5:

Line 128: please define “desaturation” based on oxygen saturation percent.

Authors reply to the Comment/Question 5:

SpO2 ≤ 93%, reported in Table 3.

  1. Reviewer Comment/Question 6:

Tables 4 and 5: Under the multivariable OR column, why are some cells blank while others have “//” present? Further, several * symbols are present in the multivariable OR column (e.g. Table 4 row Active Anticancer Treatment in Metastatic Patients, Table 5 row Type of COVID treatment”) but it is unclear where the * is directing the reader for footnotes.

Authors reply to the Comment/Question 6:

Tables 4 and 5 were modified according to Referee suggestions

  1. Reviewer Comment/Question 7:

Line 190: please define UKCCMP

Authors reply to the Comment/Question 7:

Absolutely, UKCCMP was defined as the United Kingdom (UK) Coronavirus Cancer Monitoring Project.

  1. Reviewer Comment/Question 8:

Lines 202-204: The language in the following sentence is too strong: “Furthermore, SACT de- creased the risk of mortality by 80% in patients with metastatic disease, and this remained true even after adjusting for age and gender.” This study is underpowered to draw this conclusion. Please rephrase. This is similar in the conclusion paragraph lines 254-255. Furthermore, there is high likelihood that metastatic patients not on SACT have poor performance status and/or refractory disease that precludes further SACT, which speaks to their general frailty and possibly increased vulnerability to COVID-19. If this analysis has not been performed, it should at least be mentioned in the discussion.

Authors reply to the Comment/Question 8:

Please refer to the answer to the comment/question 7 of Reviewer#1.

Reviewer 3 Report

Dear Authors,

First of all - congratulations on such good work done especially in hard times of COVID-19 pandemic. Below I present several minor and major mistakes or suggestions:

  • The abstract is well written and highlights the most crucial information, however, according to journal guidelines, the maximum length should be 200 words, whereas in this manuscript, the word count is slightly exceeded. Please correct it according to the guidelines for authors
  • Besides, the references and the way of citation should be modified according to the guidelines for authors - the brackets should be '[...]' instead of '(...)'. Please correct it.
  • Line 79 - except for the possible clinical manifestations mentioned, there were others that were reported and those concerned the nervous system, ophthalmic manifestations, or dermatological ones and this is clearly presented in a work by Baj et al. 2020. It might be probably beneficial to add this information to make it clearer for the readers.
  • Line 99 - the median and the range of patients' age is mentioned but what about the mean value? It would be beneficial to add this information.
  • Do the Authors have an information about the comorbidities - whether they were chronic or induced by SARS-CoV-2 infection in some cases? This information might be crucial here as well since it might indicate the aggressiveness of infection indicating which organs have been invaded by the virus.
  • Table 3 is quite unclear for the readers - I would recommend to change it to make it more readable.
  • Line 146 - there is a missing space which should be added after 'age'
  • Table 5 - I would recommend to make the columns broader since some of the digits are not presented in the same row.
  • Line 228 - it should be 'in' instead of 'to
  • Please check the manuscript once again in terms of English - there are several grammatical error and typos that should be corrected.

Author Response

REVIEWER#3 (Comments to the Author):

First of all - congratulations on such good work done especially in hard times of COVID-19 pandemic. Below I present several minor and major mistakes or suggestions:

  1. Reviewer Comment/Question 1:

The abstract is well written and highlights the most crucial information, however, according to journal guidelines, the maximum length should be 200 words, whereas in this manuscript, the word count is slightly exceeded. Please correct it according to the guidelines for authors. Besides, the references and the way of citation should be modified according to the guidelines for authors - the brackets should be '[...]' instead of '(...)'. Please correct it.

Authors reply to the Comment/Question 1:

Fixed

  1. Reviewer Comment/Question 2:

Line 79 - except for the possible clinical manifestations mentioned, there were others that were reported and those concerned the nervous system, ophthalmic manifestations, or dermatological ones and this is clearly presented in a work by Baj et al. 2020. It might be probably beneficial to add this information to make it clearer for the readers.

Authors reply to the Comment/Question 2:

We recognize that this information is of clinical value, though when the AIOM-L CORONA was designed the pandemic was at its first outbreak and we collected the symptoms and signs that were known at that time. Unfortunately, the requested information is not available.

  1. Reviewer Comment/Question 3:

Line 99 - the median and the range of patients' age is mentioned but what about the mean value? It would be beneficial to add this information.

Authors reply to the Comment/Question 1:

Fixed.

  1. Reviewer Comment/Question 4:

Do the Authors have an information about the comorbidities - whether they were chronic or induced by SARS-CoV-2 infection in some cases? This information might be crucial here as well since it might indicate the aggressiveness of infection indicating which organs have been invaded by the virus.

Authors reply to the Comment/Question 4:

As replied to query n.2 of Reviewer#2, comorbidities included disease or medical condition for which the patient was being treated or followed clinical instrumental controls and were defined a priori.

  1. Reviewer Comment/Question 5:

Table 3 is quite unclear for the readers - I would recommend to change it to make it more readable.

Authors reply to the Comment/Question 5:

Accordingly, we did our best to optimize the reading of Table 3.

  1. Reviewer Comment/Question 6:

Line 146 - there is a missing space which should be added after 'age'

Authors reply to the Comment/Question 6:

Fixed.

  1. Reviewer Comment/Question 7:

Table 5 - I would recommend to make the columns broader since some of the digits are not presented in the same row.

Authors reply to the Comment/Question 7:

Fixed.

  1. Reviewer Comment/Question 8:

Line 228 - it should be 'in' instead of 'to

Authors reply to the Comment/Question 8:

Fixed.

  1. Reviewer Comment/Question 9:

Please check the manuscript once again in terms of English - there are several grammatical error and typos that should be corrected.

Authors reply to the Comment/Question 9:

Done.

Round 2

Reviewer 1 Report

We thank the authors for the corrections made after the reviewers' comments.

I have only 2 minor comments for your consideration:

-Consider removing ''Despite its limitations'' from the beginning of the Conclusions section, line 323.

-The new references 31, 32 as well as 6-8 are not guidelines but consensus statements/editorial/position papers. Therefore, they should not be referred in the text as guidelines but rather as what they are.

Author Response

We thank Reviewer#1 for these additional comments. Accordingly, we replaced the term "guidelines" with "recommendations" and "expert consensus", as reported in "green" in the marked version of the text. Furthermore, as suggested, we removed ''Despite its limitations'' from the beginning of the Conclusions section.

Reviewer 3 Report

Dear Authors,

thank you for correcting the manuscript according to my previous suggestions.

All of my comments were taken into consideration and fixed accordingly.

After checking this manuscript once again with all of the comments included, I suppose that it has been significantly improved.

Thus, I have no further comments.

Best wishes with your further research.

Author Response

We thank Reviewer#3 for the nice feed back.